# Size Effect of a Piezoelectric Material as a Separator Coating Layer for Suppressing Dendritic Li Growth in Li Metal Batteries

**DOI:** 10.3390/nano13010090

**Published:** 2022-12-24

**Authors:** Junghwan Kim, Kihwan Kwon, Kwanghyun Kim, Seungmin Han, Patrick Joohyun Kim, Junghyun Choi

**Affiliations:** 1Energy Storage Materials Center, Korea Institute of Ceramic Engineering and Technology, Jinju 52851, Republic of Korea; 2Department of Applied Chemistry, Kyungpook National University, Daegu 41566, Republic of Korea

**Keywords:** Li metal battery, separator modification, piezoelectric material, size effect

## Abstract

Li metal has been intensively investigated as a next-generation rechargeable battery anode. However, its practical application as the anode material is hindered by the deposition of dendritic Li. To suppress dendritic Li growth, introducing a modified separator is considered an effective strategy since it promotes a uniform Li ion flux and strengthens thermal and mechanical stability. Herein, we present a strategy for the surface modification of separator, which involves coating the separator with a piezoelectric material (PM). The PM-coated separator shows higher thermal resistance than the pristine separator, and its modified surface properties enable the homogeneous regulation of the Li-ion flux when the separator is punctured by Li dendrite. Furthermore, PM was synthesized in different solvents via solvothermal method to explore the size effect. This strategy would be helpful to overcome the intrinsic Li metal anode problems.

## 1. Introduction

In the current context of environmental problems stemming from global warming caused by excessive greenhouse gas emissions, achieving carbon neutrality is becoming an urgent goal [1,2,3]. CO_2_ emitted by the private vehicles, which uses fossil fuels such as petrol and coal, accounts for 77% of the total greenhouse gas emissions [4,5]. Hence, the trends of the mobile vehicle are shifting from internal combustion engine vehicles to electric vehicles (EVs) [6,7,8]. Li ion batteries (LIBs) are commercially used as energy storage devices for EVs owing to their good cyclability, safety, and rate capability [7,9,10]. Unfortunately, the energy density of the currently available commercial LIBs is limited to less than 250 Wh kg^−1^, which is insufficient to provide a decent driving range for EVs [8,11]. Therefore, increasing the energy density of batteries is demanded.

Li metal has a high specific capacity (3860 mAh g^−1^) and the lowest reduction potential (−3.04 V versus the standard hydrogen electrode at room temperature). These advantages render it the ultimate candidate as a next-generation rechargeable battery anode material [11,12,13,14,15,16,17,18]. However, Li metal suffers from a huge volume change during repeated Li deposition/dissolution processes [8,13]. This nature results in the breakdown/restoration of the fragile solid electrolyte interphase (SEI) layer over the surface of the Li metal anode [15,16,19]. Consequently, during Li deposition, Li ion flux is locally intensified at the cracks of the SEI layer, leading to the formation of Li dendrites [15]. Sharp tips of Li dendrites can penetrate the separator, which causes safety issues (e.g., thermal runaway and explosion hazards) [19]. Moreover, isolated Li and porous electrode can be formed during repeated Li deposition/dissolution processes. These phenomena lead to the degradation of the electrochemical performance in terms of Coulombic efficiency and capacity [8]. Due to these problems, Li metal was disregarded as an anode material for ~40 years [8,20]. To overcome the limitations of the LMB technology, countless strategies have been adopted, such as optimizing the electrolytes for a robust SEI, directly constructing an artificial SEI, and using three-dimensional current collectors [17,21,22,23,24,25,26,27,28,29]. These strategies, which mainly aim at stabilizing the SEI layer or reducing the effective current density on the Li anode, have successfully increased the safety and cycle performance of LMBs. Unfortunately, these approaches involve complicated processes and harsh conditions, which restrict the practical applications of LMBs. 

Among all the strategies, the introduction of functional separators has emerged as one of the most efficient strategies for enhancing LMB performance. The separator performs three essential functions, i.e., providing an ion path, serving as an electrolyte reservoir, and serving as a physical barrier between anode and cathode [12,30,31,32,33]. Considering the roles of the separator, separator modification can be regarded as a promising strategy for enhancing the cycle performance and for solving the safety issues of LMBs by inducing a uniform Li ion flux and by improving thermal/mechanical stability. Therefore, many attempts have been devoted to making the LMBs feasible by depositing various functional materials on the separator. However, further research is still required to achieve long cycle life and to reduce parasitic reactions occurring in LMBs [12,19].

A BaTiO_3_ that exhibits both ferroelectricity and piezoelectricity has a relatively high Curie temperature (Tc = 120 °C) and possesses a high dielectric constant (≥ 1500 at RT) and low dielectric loss [34]. In this study, we modify the surface of a separator by coating it with a piezoelectric material (PM). The PM was synthesized via a solvothermal method. Owing to its property of generating an electric field in response to an applied mechanical stress [35,36,37], the PM, which was coated on the commercial separator using a hydrophilic polymer binder, promotes a homogeneous Li ion flux when the PM-coated separator is punctured by dendritic Li [38]. This leads to enhanced electrochemical characteristics compared with those of the pristine polypropylene (PP) separator. Furthermore, to investigate the size effect of PM on the LMBs, PMs with different sizes were synthesized. This study provides a new possibility for the modification of separators.

## 2. Materials and Methods

### 2.1. Raw Materials 

The following raw materials were used for synthesizing barium titanate (BaTiO_3_) as a PM: barium hydroxide octahydrate (Ba(OH)_2_·_8_H_2_O; ≥ 98% purity, Sigma-Aldrich, St. Louis, MO, USA), titanium(IV) oxide (anatase, TiO_2_; ≥ 99% purity, Sigma-Aldrich, St. Louis, MO, USA), 1-butanol (C_4_H_10_O; 99.9% purity, Sigma-Aldrich, St. Louis, MO, USA), and ethyl alcohol (C_2_H_6_O; 99.5% purity, SAMCHUN, Gangnam-gu, Seoul, Republic of Korea).

### 2.2. Synthesis of the Different Sized PM

A large and small PM (LPM and SPM, respectively) were synthesized via a solvothermal method using deionized water (DIW) and 1-butanol as solvents, respectively. First, 20 mmol of Ba(OH)_2_·_8_H_2_O and 10 mmol of TiO_2_ were stirred in a 40 mL Teflon reactor for 5 min at 350 rpm. Then, the Teflon reactor was set in a stainless steel autoclave and heated at 200 °C for 72 h. After the autoclave was cooled to room temperature, the product was centrifuged at 10,000 rpm and washed with water and ethanol successively for three times each. The rinsed product was dried in an oven at 60 °C. To remove the second phase, acetic acid treatment was conducted. Specifically, 2 g of the product was mixed with 50 mL of an acetic acid aqueous solution (0.69 mol dm^−3^) and stirred for 5 min at 350 rpm. The rinsing and drying process was repeated to obtain the product.

### 2.3. Preparation of PM-Coated Separator

A PM slurry was prepared by mixing 90% PM, polyacrylic acid (Mv ~450,000, Sigma-Aldrich, St.Louis, MO, USA), and DIW with a planetary mixer (ARE-310, THINKY). This slurry was laminated onto a PP separator (Celgard 3501, CELGARD, Charlotte, NC, USA) through tape-casting. The coated separator was dried overnight in a vacuum oven at 60 °C. The thickness of the coating layer was 10 µm.

### 2.4. Characterization

The morphologies of LPM and SPM were investigated through field emission scanning electron microscopy (FE-SEM, MIRA II LMH, TESCAN, Brno-Kohoutovice, Czech Republic). The particle-size distributions of LPM and SPM were analyzed by using a particle size analysis (PSA, LA-950V2, Horiba, Kyoto, Japan). X-ray diffraction (XRD, D8 Advance, Bruker, Billerica, MA, USA) was performed to confirm the crystallinity of PM.

### 2.5. Measurement of Electrochemcial Performance

To investigate the effect of PM on the electrochemical stability, Li metal foil was paired with Cu foil in a coin cell configuration (2032 R type). The Li metal foil was attached with a stainless spacer with an additive-free carbonate-based electrolyte (1.0 M lithium hexafluorophosphate (LiPF6) in a mixture of ethylene carbonate and diethyl carbonate (EC:DEC = 1:1 vol.%)). Each separator was punched to a diameter of 18 mm using a punching cutter before cell assembly. The Li deposition/dissolution measurements were performed at 1.0 mAh cm^−2^. The area capacity of the Li deposition process was 1 mAh cm^−2^, and the cutoff voltage of Li stripping was 1.0 V. Electrochemical impedance spectroscopy (EIS, VSP, Biologic, Seyssinet-Pariset, France) was performed in the frequency range of 250 kHz–10 MHz. Nyquist plot for EIS was fitted by equivalent circuit.

## 3. Results

Figure 1 shows the mechanism of dendritic growth suppression induced by the piezoelectric properties of PM. When Li protrusion occurs during the cycling process, the current density is localized near the tip of the Li metal. The localized stress fractures the SEI layer, which accelerates the growth of dendritic Li. This event eventually penetrates the pristine PP separator and causes safety issues such as battery short circuits. In contrast, the PM-coated separator mitigates these problems. The piezoelectric effect is the ability of a polarized material to generate an electric field in response to an applied physical stress. When dendritic Li applies physical pressure to the PM-coated separator, the PM is polarized, resulting in the generation of an electric field in the opposite direction to that of Li deposition. This opposite electric field regulates the Li-ion flux, thereby reducing the local current density and inducing a uniform Li deposition [35,39]. Moreover, the PM coating layer physically prevents the dendritic Li growth and enhances thermal stability. Hence, the PM-coated separator can help achieve a stable cycle performance.

The particle morphologies of LPM and SPM were characterized by scanning electron microscope (SEM). Figure 2a–c show the particle size distribution measured by PSA and representative SEM images of PMs synthesized via a solvothermal method using different solvents. The size distributions of LPM and SPM were 1–5 µm and 70–100 nm, respectively. This implies that SPM is suitable for the generation of a uniform electric field owing to its high particle number density per unit area. The nanoscale design of SPM would polarize the surface of separator more uniformly and generate a dense electric field. Both LPM and SPM exhibited a cubic structure with a smooth surface, which is conducive to preventing damage to the separator by a physical impact. An X-ray diffraction (XRD) analysis was conducted to verify the presence of impurities in the PM (Appendix A). The XRD results indicate that both of LPM and SPM have typical tetragonal crystal structures of BaTiO3 (JCPDS No. 5-0626) without any impurity phase [40].

Figure 2d–f display the surface morphologies of the pristine PP and the PM-coated separators. The SEM image of the pristine PP separator showed a plane surface with submicron pores over the entire separator. In contrast, a uniform deposition of PM particles covering the porous surface was observed after coating with a PM layer.

A contact-angle analysis was performed to compare the electrolyte wettability of each separator (Figure 3a). The same amount of polar solvent, i.e., water, was dropped onto each separator to estimate the contact angle. The LPM- and SPM-coated separators exhibited a lower contact angle (71° and 65°, respectively) than that (95°) of the pristine PP separator. This result indicates that the PM-coating layer improves polar solvent affinity and wettability [41]. Thermal shrinkage of the separator is a significant risk affecting battery safety [42]. Therefore, samples (3 cm × 3 cm) were subjected to thermal treatment to evaluate thermal shrinkage. Figure 3b shows the images of the pristine PP separator, and the LPM- and SPM-coated separators before and after thermal treatment at 100 °C, 120 °C, 140 °C, and 160 °C for 1 h. No differences between the separators were observed up to 100 °C. However, the thermal shrinkage of the pristine PP separator was accelerated above 120 °C due to the low thermal stability of PP [43]. In contrast, the introduction of PM diminished the thermal shrinkage of the pristine PP separator since the PM coating layer enhanced the mechanical resistance of the entire separator, including the tensile and puncture strengths [44,45,46]. These results are summarized in Figure 3c. The improved thermal property is beneficial in preventing thermal runaway.

To compare the electrochemical stabilities of cells using different separators, the Li deposition/dissolution tests were conducted. Figure 4a shows the cycle performance of the Li-Cu cells at 1.0 mA cm^−2^. The Coulombic efficiencies (CEs) of the cells with the pristine PP separator and LPM-coated separator dropped below 80% after 80 cycles, due to severe dendritic growth which causes dead Li and induces high internal resistance [47]. Furthermore, the cell with the SPM-coated separator maintained a high CE (90%) even at 150 cycles. This result indicates that nanosized SPM effectively suppresses the growth of dendritic Li by constructing a dense electric field that disperses the local current density concentrated on the Li metal tips [35]. The ceramic coating layer also serves as a mechanical barrier to prevent sharp Li dendrites from damaging the separator, which results in stable cycle performance. Figure 4b1–b3 show the voltage profiles of each Li-Cu cell at the 1st, 50th, and 150 cycles. At the initial nucleation step, the average nucleation overpotentials of Li at the cells with LPM- and SPM-coated separators are slightly higher than that of the cell with the pristine PP separator due to the covering of the pores by the PM coating layer. However, upon Li dendrite growth and the generation of a homogeneous electric field, the PM-coated separator reduced the resistance of Li deposition. Moreover, the cell with the SPM-coated separator presents clearer and longer discharge region than the cells with the pristine PP separator and LPM-coated separator at 150 cycles. These results mean that the dual role of the SPM coating layer (constructing a dense electric field and physically protecting separator) contributes to the superior electrochemical stability by the suppression of dendritic Li.

EIS measurements were conducted to confirm the difference in cycle stability. Since the PM coating layer covers the pores, as shown in Figure 2d–f, the LPM- and SPM-coated separators exhibit relatively lower ionic conductivities (1.32 and 1.37 mS cm^−1^, respectively) than the pristine PP separator (1.61 mS cm^−1^) (Figure 5a). The low ionic conductivities of the cells with PM-coated separators correspond with the nucleation overpotential shown in Figure 4b1. Since the PM coating layer physically covers the pores of the separator, the resistance of the cells with LPM- and SPM-coated separators slightly increased. Nevertheless, the internal resistance of the cells with the PM-coated separator was improved after polarization. The charge-transfer resistance values of the cell with the pristine PP separator, and the LPM- and SPM-coated separators were 362.2, 211.3, and 175.6 Ω, respectively, after one cycle (Figure 5b). The cell with the pristine PP separator showed the highest total resistance owing to the growth of dendritic/mossy Li metal. In contrast, under the same situations, the cells with the PM-coated separators exhibited lower charge-transfer resistance, indicating that the growth of dendritic Li was controlled through uniform Li ion flux. In addition, considering that the cell with SPM-coated separator showed the lowest resistance, it is likely that the SPM-coated separator is capable of constructing a denser electric field than the LPM-coated separator.

The cycled Li-Cu cells at 150 cycles were disassembled to examine the cycle stability. Figure 6 shows representative SEM images of the cycled Cu current collectors with the pristine PP separator, and LPM- and SPM-coated separators. The surface of the current collector with the pristine PP separator showed long needle-like clusters of Li dendrite formed by unstable Li deposition/dissolution processes (Figure 6a). In the case of the current collector with the LPM-coated separator, the LPM coating layer physically suppressed Li dendrite growth. However, compact clusters of Li dendrite with relatively large diameters were still observed (Figure 6b). Conversely, a flat surface of Li metal was observed in the current collector with the SPM-coated separator (Figure 6c), indicating that uniform Li deposition occurred as a result of the dense electric field. Therefore, it can be concluded that SPM comprising nanosized particles can enhance the cycle stability of LMBs.

## 4. Conclusions

In this study, we report an effective strategy for separator modification via PM coating. The PM-coated separators exhibit excellent thermal resistance compared to the pristine PP separator. In terms of the electrochemical properties, PM-coated separator plays two important roles: (1) the formation of a piezoelectric field and (2) serving as a physical barrier to suppress the Li dendrite growth. Additionally, the concept of the PM coating layer is effectively realized at the nanosized scale. The SPM coating layer having nanosized particles generates a dense electric field that enables a uniform Li deposition. Thus, the SPM-coated separator showed better electrochemical performance than the other separators, resulting in superior Coulombic efficiency during 150 cycles. Considering these results, our proposed approach to separator modification would be an efficient strategy to solve the intrinsic problems in practical LMBs.

## Figures and Tables

**Figure 1 nanomaterials-13-00090-f001:**
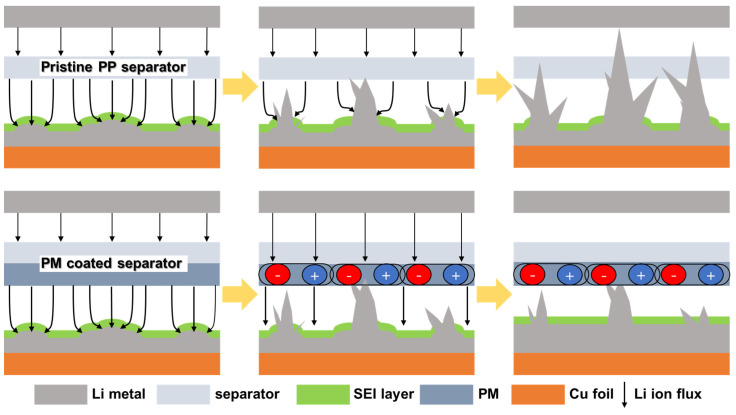
Schematic illustration of the polarization process occurring in the pristine PP separator (**top**) and the PM-coated separator (**bottom**).

**Figure 2 nanomaterials-13-00090-f002:**
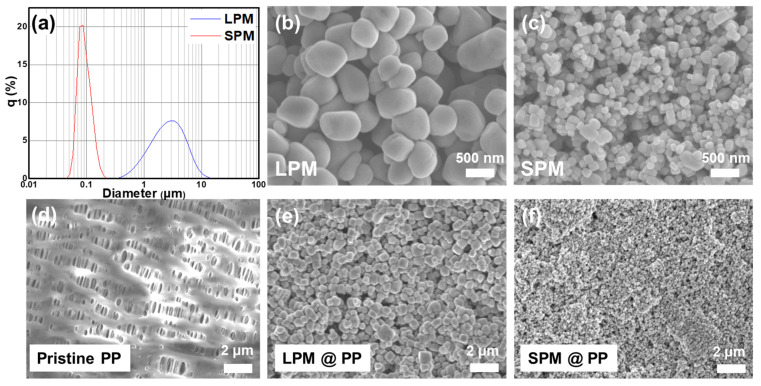
(**a**) Particle size distribution of LPM and SPM; (**b**,**c**) SEM images of LPM and SPM, respectively; (**d**–**f**) SEM images of the pristine PP separator, and the LPM- and SPM-coated separators, respectively.

**Figure 3 nanomaterials-13-00090-f003:**
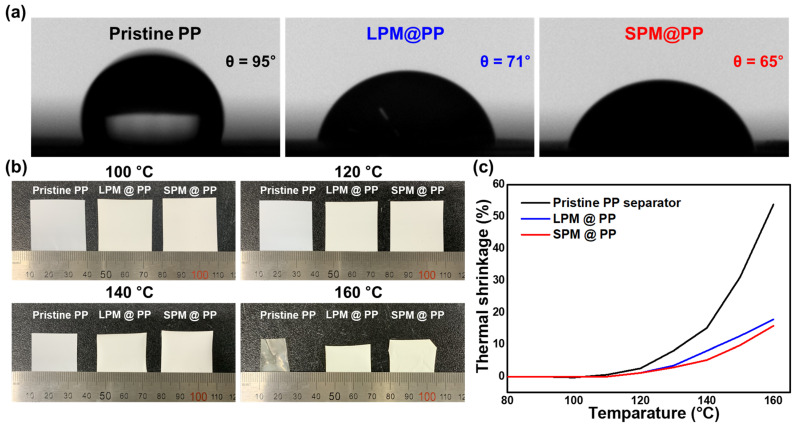
(**a**) Contact-angle measurements using a pristine PP separator, and the LPM- and SPM-coated separators. (**b**) Thermal shrinkage photographs at 100 °C, 120 °C, 140 °C and 160 °C; (**c**) Thermal shrinkage percentage of pristine of the pristine PP separator, and the LPM- and SPM-coated separators.

**Figure 4 nanomaterials-13-00090-f004:**
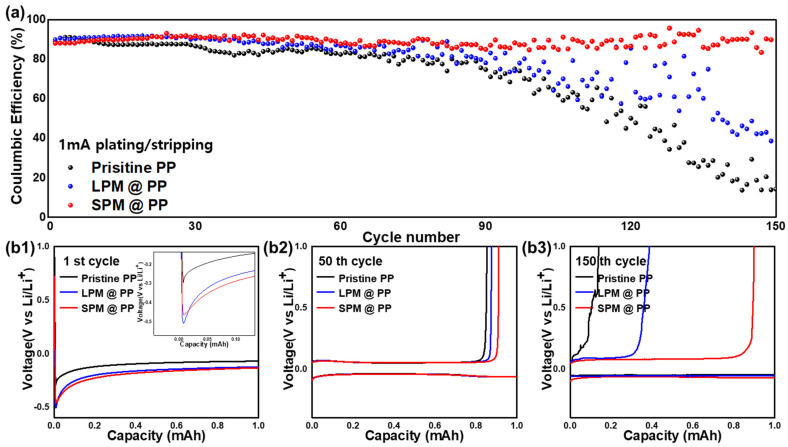
Electrochemical performances of Li-Cu cells with the pristine PP separator, and the LPM- and SPM-coated separators. (**a**) Coulombic efficiency at a current density of 1 mA cm^−1^ and (**b1**–**b3**) voltage profiles at the 1st, 50th, and 150th cycles.

**Figure 5 nanomaterials-13-00090-f005:**
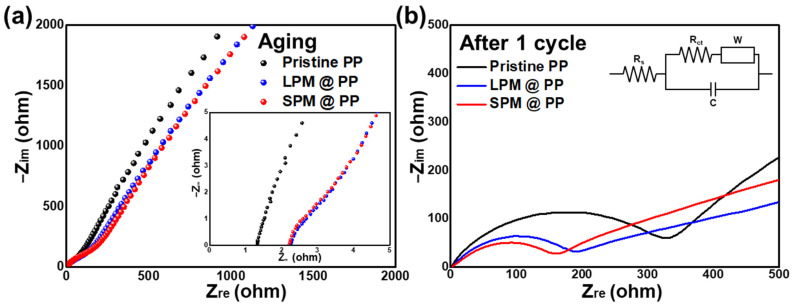
Electrochemical impedance spectra (EIS) of Li-Cu cells with pristine PP separator, and the LPM- and SPM-coated separators (**a**) before and (**b**) after one cycle.

**Figure 6 nanomaterials-13-00090-f006:**
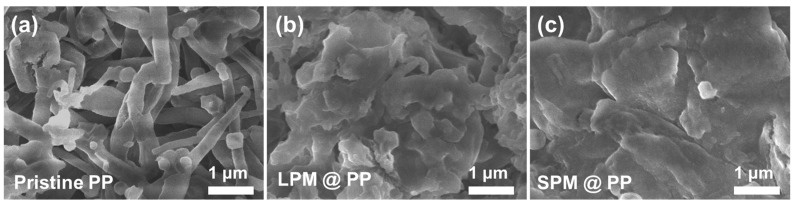
Post-mortem examinations after cycling. SEM images of the cell with (**a**) pristine PP separator, (**b**) LPM-coated separator, and (**c**) SPM-coated separator.

## Data Availability

The data presented in this study are available on request from the corresponding author. The data are not publicly available.

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
