# Peer review of "Size Effect of a Piezoelectric Material as a Separator Coating Layer for Suppressing Dendritic Li Growth in Li Metal Batteries"

_nanomaterials, 2022, doi:10.3390/nano13010090_

Round 1
Reviewer 1 Report
In the work "Size effect of a piezoelectric material as a separator coating layer for suppressing dendritic Li growth in Li metal batteries" authors considered the effect of a piezoelectric layer of barium titanate on the formation of lithium dendrites in a battery with an electrode of lithium metal. The authors showed that cells with a separator layer coated with submicron BaTIO3 demonstrate inhibition of dendritic growth, resulting in greater stability of the cell.
Questions:
1) The authors claim that coulombic efficiencies of the cells with the pristine separator dropped dramatically due to severe dendritic growth. It is not clear why this effect is the cause.
2) The authors do not confirm the proposed mechanism for suppressing the formation of dendrites due to the piezoelectric effect of standing BaTiO3. It is possible that the barium titanate layer acts as a mechanical barrier.
3) The difference in EIS results is not clear. The impedance spectra must be described by a satisfactory equivalent circuit, after which they can be compared with each other.
Author Response
Editor and reviewer comments
Reviewer: 1
Overall Assessment: In the work "Size effect of a piezoelectric material as a separator coating layer for suppressing dendritic Li growth in Li metal batteries" authors considered the effect of a piezoelectric layer of barium titanate on the formation of lithium dendrites in a battery with an electrode of lithium metal. The authors showed that cells with a separator layer coated with submicron BaTIO3 demonstrate inhibition of dendritic growth, resulting in greater stability of the cell.
Our response: We thank the reviewer for the overall assessment on our manuscript.
Comment 1. The authors claim that coulombic efficiencies of the cells with the pristine separator dropped dramatically due to severe dendritic growth. It is not clear why this effect is the cause.
Our response: We thank the reviewers for pointing out the lack of explanation. The dendritic growth promotes the generation of dead Li which induces high resistance. As the reviewer pointed out, we added a description why severe dendritic growth causes negative effect on coulombic efficiency. This description is mentioned in Figure 4 section and related reference was added. We again appreciate the valuable comment.
Comment 2.The authors do not confirm the proposed mechanism for suppressing the formation of dendrites due to the piezoelectric effect of standing BaTiO3. It is possible that the barium titanate layer acts as a mechanical barrier.
Our response: We thank the reviewer for the comments on our manuscript. The BaTiO3 coating layer not only serves as a mechanical barrier but also distributes the local current density with a piezoelectric effect. This can be disproved through the results of Figure 4a. The cell with the SPM-coated separator prevented short circuit from sharp Li metal tip (mechanical barrier) and improved the cycle performances by suppressing the growth of dendrite (uniform local current density by piezoelectric effect). To avoid the possible confusion by readers, we’ve added the reference related to piezoelectric effect and indicated the information “mechanical barrier”. We again appreciate the valuable comment.
Comment 3. The difference in EIS results is not clear. The impedance spectra must be described by a satisfactory equivalent circuit, after which they can be compared with each other.
Our response:We thank the reviewers for pointing out the details. As advised by the reviewer, we added a circuit model for Nyquist plots at Figure 5b. And we mentioned the existence of these circuit models in the Measurement of Electrochemical Performance part in Materials and Method section. This information would be very helpful for other researchers to refer to this experiment.

Reviewer 2 Report
In this work, a piezoelectric material (PM) coated separator shows higher thermal resistance than that of the pristine separator, and its modified surface properties enable the homogeneous regulation of the Li-ion flux when the separator is punctured by Li dendrite. This strategy would be helpful to overcome the intrinsic Li metal anode problems. However, there are a few issues that should be addressed before considering its publication. These questions are listed as follows:
1. What are the advantages of barium titanate (BaTiO3) over other piezoelectric materials?
2. The electrolyte wettability of a separator is the key to the electrochemical performance of LIBs. What is the electrolyte wettability of the pristine PP separator, the LPM- and SPM-coated separators?
3. What is the effect of piezoelectric material on the tensile properties of the pristine PP separator? Please provide the stress-strain curves for the pristine PP separator, the LPM- and SPM-coated separators.
4. Please provide long-term cycling performance of the full cells assembled with the pristine PP separator, the LPM- and SPM-coated separators.
5. Please provide rate performance of the full cells assembled with the pristine PP separator, the LPM- and SPM-coated separators.
Author Response
Editor and reviewer comments
Reviewer: 2
Overall Assessment :In this work, a piezoelectric material (PM) coated separator shows higher thermal resistance than that of the pristine separator, and its modified surface properties enable the homogeneous regulation of the Li-ion flux when the separator is punctured by Li dendrite. This strategy would be helpful to overcome the intrinsic Li metal anode problems. However, there are a few issues that should be addressed before considering its publication. These questions are listed as follows:
Our response :We thank the reviewer for the overall assessment on our manuscript.
Comment 1. What are the advantages of barium titanate (BaTiO3) over other piezoelectric materials?
Our response:We thank the reviewers for the comments on our manuscript. As the reviewer pointed out, the explanation was omitted. Therefore, we added a explanation why the barium titanate (BaTiO3) is better than other piezoelectric materials. BaTiO3 possesses a high dielectric constant at room temperature and low dielectric loss. In addition, the Curie temperature of barium titanate is relatively high so that barium titanate can be used at temperature as high as 120 °C. Due to these properties, among materials with piezoelectric effect, BaTiO3 was considered appropriate for our study.
Comment 2.The electrolyte wettability of a separator is the key to the electrochemical performance of LIBs. What is the electrolyte wettability of the pristine PP separator, the LPM- and SPM-coated separators?
Our response:We thank the reviewer for the comments on our manuscript. As reviewer commented, we have added the electrolyte wettability information in Figure 3. And we also additionally explained about this information. This information would be very helpful for other researchers to refer to this experiment.
Comment 3.What is the effect of piezoelectric material on the tensile properties of the pristine PP separator? Please provide the stress-strain curves for the pristine PP separator, the LPM- and SPM-coated separators.
Our response: We thank the fruitful comment from the reviewer. In the previous report, the coating of ceramic materials was associated with increased tensile strength. (such as ACS Applied Materials & Interfaces 2015, 7, 24119-24126 and Advanced Functional Materials 2016, 26, 7817-7823). As reviewer commented, the stress-strain curve is a good indicator of the physical properties of each sample. To compensate this parameter, we plan to conduct further studies including the tensile strength test in a near future work.
Comment 4. Please provide long-term cycling performance of the full cells assembled with the pristine PP separator, the LPM- and SPM-coated separators.
Our response:We appreciate the fruitful comment from the reviewer. In the cycling performance of the half cells, the coulombic efficiencies (CEs) of the cells with the pristine PP separator and LPM-coated separator dropped below 80 % after 80 cycles. Meanwhile, the cell with the SPM-coated separator maintained a high CE (90 %) even at 150 cycles due to the effective suppression of the dendrites. Through the electrochemical performance of the half-cell, it can be inferred that full cell data will also have the same tendency. As reviewer recommended, we plan to conduct further studies including the full-cell test in a near future work.
Comment 5. Please provide rate performance of the full cells assembled with the pristine PP separator, the LPM- and SPM-coated separators.
Our response: We thank the fruitful comment from the reviewer. As shown in Figure 5b, the cells with the LPM- and SPM-coated separator exhibited improved charge-transfer resistance (211.3 and 175.6 Ω, respectively) compared to the cell with the pristine PP separator (362.2 Ω). Judging from the enhanced electrical property of the cells with the PM-coated separators, it is considered that the PM-coating layer is also effective for rate performance. As reviewer suggested, the results of rate performances would be helpful for reader’s understanding. To come up to the reviewer’s expectation and do something meaningful for the battery society, we are planning to perform more studies suggested by the reviewer in a near future work.

Round 2
Reviewer 1 Report
The article was slightly improved and the authors answered questions. I think that the manuscript can be accepted for publication.
Reviewer 2 Report
The revised manuscript can be accepted.